# Adapter-TST: A Parameter Efficient Method for Multiple-Attribute Text Style Transfer

**Zhiqiang Hu**[1]   **Roy Ka-Wei Lee**[1]   **Nancy F. Chen**[2]
[1]Singapore University of Technology and Design, Singapore
[2]Institute of Infocomm Research (I2R), A*STAR, Singapore

## Abstract

Adapting a large language model for multiple-attribute text style transfer via fine-tuning can be challenging due to the substantial amount of computational resources and labeled data required for the specific downstream task. In this paper, we address this challenge by introducing Adapter-TST, a framework that freezes the pre-trained model's original parameters and enables the development of a multiple-attribute text style transfer model. Using BART or T5 as the backbone model, Adapter-TST utilizes different neural adapters to model different types of attribute information, similar to a plug-in connected to the base model. Our method allows control over multiple attributes (e.g. sentiment, tense, active or passive voice) and configures the adapters' architecture to generate multiple outputs in respect to attributes or compositional editing on the same sentence. We evaluate the proposed model on both traditional sentiment transfer and multiple-attribute transfer tasks. The experiment results demonstrate that Adapter-TST outperforms all the state-of-the-art baselines with significantly less computational resources. We have also empirically shown that each adapter is able to characterize specific stylistic attributes effectively and can be configured to perform compositional editing. The code and datasets can be found in https://github.com/Social-AI-Studio/Adapter-TST.

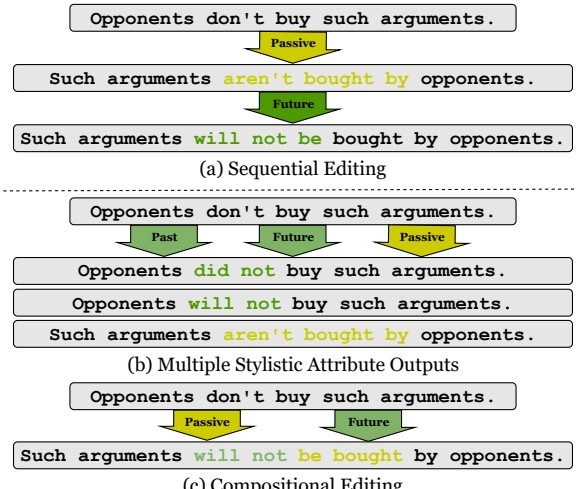

(a) Sequential Editing

(b) Multiple Stylistic Attribute Outputs

(c) Compositional Editing

Figure 1: Examples of different settings of multiple-attribute TST. (a) Existing single-attribute TST models perform sequential editing by transferring the text style sequentially to achieve compositional editing. Multiple-attribute TST models can (b) generate multiple outputs simultaneously in the corresponding target style, or (c) perform compositional editing by transferring different target styles. The proposed Adapter-TST enables a single PLM to achieve both settings (b) and (c) by configuring the adapters' connection method.

## 1   Introduction

**Motivation.** Text style transfer (TST) is a popular natural language generation task that aims to change the stylistic properties (e.g., sentiment, formality, tense, voice) of the text while preserving the style-independent content (Hu et al., 2022a). Existing studies explore performing text style transfer on attributes like age, or gender (Lample et al., 2019), sentiment (Li et al., 2018; Luo et al., 2019; Fu et al., 2018), formality (Rao and Tetreault, 2018), politeness (Madaan et al., 2020; Hu et al., 2022b), and author writing style (Syed et al., 2020). Nevertheless, most of the existing TST studies are confined to single-attribute TST tasks.

Few works have explored multiple-attribute TST tasks, where TST models are designed to control and transfer text in multiple target stylistic attributes. Lample et al. (2019) attempts style transfer with multiple attributes by conditioning on the average embedding of each target attribute and using a combination of denoising autoencoder (DAE) and back-translation techniques. Goyal et al. (2021) propose an approach to initialize an encoder-decoder setup with a transformer-based language model that is pre-trained on a generic corpus and enhances its capability of re-writing to multiple target style dimensions by utilizing multiple style-

aware language models as discriminators.

A possible approach to perform single and multiple attribute TST tasks is to leverage large pre-trained language models (PLMs). The PLMs have been pre-trained on large corpora, which allows them to capture natural language's syntactic and semantic information. This characteristic of PLMs makes them well-suited for TST tasks, where the model needs to understand the content and style of the input text. Syed et al. (2020) fine-tune a denoising autoencoder (DAE) for the stylized re-writing task by initializing the encoder and decoder with a pre-trained language model trained on Masked Language Modeling (MLM) objectives (Devlin et al., 2019). Wang et al. (2019) fine-tune GPT-2 model (Radford et al., 2019) using the text formality transfer rules harnessed from analyzing the GYAFC parallel dataset (Rao and Tetreault, 2018). The fine-tuned GPT-2 model was subsequently used to transfer the formality of text (e.g., informal to formal text). However, fine-tuning PLMs for multiple-attribute TST remains challenging as a significant amount of computational resources and style-labeled data are required to perform TST for each stylistic attribute.

**Research Objectives.** To address these research gaps, we propose Adapter-TST, a parameter-efficient framework that utilizes BART (Lewis et al., 2020) or T5 (Raffel et al., 2020) as the backbone model and trains neural adapters to capture multiple stylistic attributes for multiple-attribute TST. During the training of Adapter-TST, we freeze the original parameters of the pre-trained BART or T5 model and only update the parameters of adapters to relax the dependence on computational resources and supervised data. The proposed Adapter-TST model is flexible to handle different settings of multiple-attribute TST by configuring the connection method among adapters. Figure 1 illustrates the different settings of multiple-attribute TST tasks. Paralleling the adapters in Adapter-TST can generate multiple outputs in the corresponding target style simultaneously (setting b) and stacking the adapters for compositional editing in terms of different target styles at the same time (setting c). We conduct experiments on the traditional sentiment transfer task and multiple-attribute TST tasks, including multiple stylistic attribute outputs and compositional editing. Results of automatic and human evaluations show that Adapter-TST can outperform the state-of-the-art baselines

to transfer and generate high-quality text with lower computational resources.

**Contributions.** We summarize our contributions as follows: (i) We introduce an Adapter-TST, which is a parameter-efficient framework that can perform multiple-attribute TST tasks with significantly lower computational resources. (ii) Included in the Adapter-TST are two TST configurations, *parallel* and *stacking*, which support multiple-output TST and compositional editing, respectively. (iii) We conducted extensive experiments on real-world datasets. The automatic and human evaluation results show that Adapter-TST can outperform the state-of-the-art baselines to transfer and generate high-quality text.

## 2 Related Work

### 2.1 Text Style Transfer

TST is an emerging research topic that has garnered attention from computational linguists and computer science researchers. The recent comprehensive survey (Hu et al., 2022a; Jin et al., 2022) summarizes the existing TST approaches.

While the majority of existing studies have focused on performing TST on single attributes such as sentiment (Li et al., 2018; Luo et al., 2019; Fu et al., 2018) or formality (Rao and Tetreault, 2018), recent studies have also explored multiple-attribute TST tasks, where TST models are designed to control and transfer text in multiple target stylistic attributes. Lample et al. (2019) attempts style transfer with multiple attributes by conditioning on the average embedding of each target attribute and using a combination of denoising autoencoder (DAE) and back-translation techniques. Goyal et al. (2021) propose an approach to initialize an encoder-decoder setup with a transformer-based language model that is pre-trained on a generic corpus and enhances its capability of re-writing to multiple target style dimensions by utilizing multiple style-aware language models as discriminators. In this study, we contribute to this limited multiple-attribute TST literature by proposing an alternative approach to generate multiple stylistic outputs and perform compositional editing efficiently.

Due to the lack of parallel training data, most existing TST methods are designed to train with non-parallel style-labeled sentences as input. A popular line of TST approaches aims to disentangle the text's content and style in the latent space to perform TST (Shen et al., 2017; Zhao et al., 2018;

Fu et al., 2018; Chen et al., 2018; Logeswaran et al., 2018; Yin et al., 2019; Lai et al., 2019; Vineet et al., 2019). Another common approach is to leverage PLMs. For instance, Syed et al. (2020) fine-tune a denoising autoencoder (DAE) for the stylized re-writing task by initializing the encoder and decoder with a pre-trained language model trained on Masked Language Modeling (MLM) objectives (Devlin et al., 2019). Wang et al. (2019) fine-tune GPT-2 model (Radford et al., 2019) using the text formality transfer rules harnessed from analyzing the GYAFC parallel dataset (Rao and Tetreault, 2018). The fine-tuned GPT-2 model was subsequently used to transfer the formality of text (e.g., informal to formal text). However, fine-tuning PLMs for multiple-attribute TST remains challenging as a significant amount of computational resources is required to perform the task; multiple PLMs need to be fine-tuned for the different attributes to perform multiple-attribute TST. In this study, we overcome this limitation by proposing Adapter-TST, which is a parameter-efficient framework that leverages on PLMs but requires significantly lesser computational resources to perform multiple-attribute TST.

## 2.2 Adapter-based Models

PLMs, pre-trained on large-scale text corpus with unsupervised objectives, have established state-of-the-art performances on various NLP downstream tasks. Many studies fine-tune PLMs with language modeling and downstream task objectives to obtain better performance (Zhang et al., 2019b; Lauscher et al., 2019; He et al., 2019; Xiong et al., 2019).To leverage the powerful PLMs more efficiently, Houlsby et al. (2019) add adapter layers, small neural networks, into each transformer layer to obtain near state-of-the-art performance on the GLUE benchmark while updating only the parameters of adapters. Inspired by this work, more adapter-based models (Wang et al., 2021; Liu et al., 2021; Zhong et al., 2021) are proposed to inject task-specific knowledge into PLMs with adapters. Inspired by the adapter architecture, we propose Adapter-TST, which trains different neural adapters to capture different stylistic attributes to perform the multiple-attribute TST. The proposed adapter framework has two configurations that support multiple stylistic attribute outputs and compositional editing.

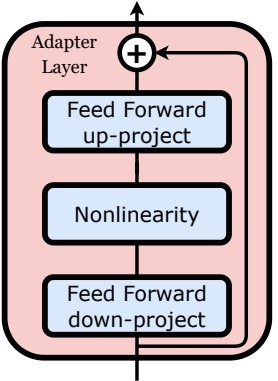

Figure 2: Structure of the adapter layer. The adapter layer consists of a bottleneck with up and down projection layers, and a skip connection between two projection layer.

## 3 Methodology

This section proposes Adapter-TST, which adds neural adapters into each transformer layer to capture different attribute information for multiple-attribute TST. We first introduce the adapter structure used in Adapter-TST and its parameter efficiency. Subsequently, we explain how the adapters are configured for different multiple-attribute TST settings, namely, *multiple stylistic attribute outputs* and *compositional editing*. Finally, we describe the training objectives of Adapter-TST.

### 3.1 Adapter Structure

We present an adapter structure in Figure 2. The adapter consists of a bottleneck that contains few parameters relative to the attention and feedforward layers in the original model. A skip connection is applied across two projection layers. In our proposed Adapter-TST, these adapters will be trained to capture different stylistic attributes. In contrast to Houlsby et al. (2019), adding the adapter module twice to each transformer layer, we propose simplifying the approach by just adding the adapter layer into each transformer once, making our Adapter-TST's architecture more parameter efficient.

We use BART-large (24-layer, 1024-hidden, 16-heads, 406M parameters) or T5-large (24-layer, 1024-hidden, 16-heads, 770M parameters) as the backbone model in Adapter-TST. As for each adapter layer, we denote the hidden dimensions of the down-projection and up-projection layers as $H_d = 64$ and $H_u = 1024$. The bottleneck adapter layers are plugged into each layer of BART-large or T5-large, and different adapter layers do not share parameters. Thus the total number of parameters

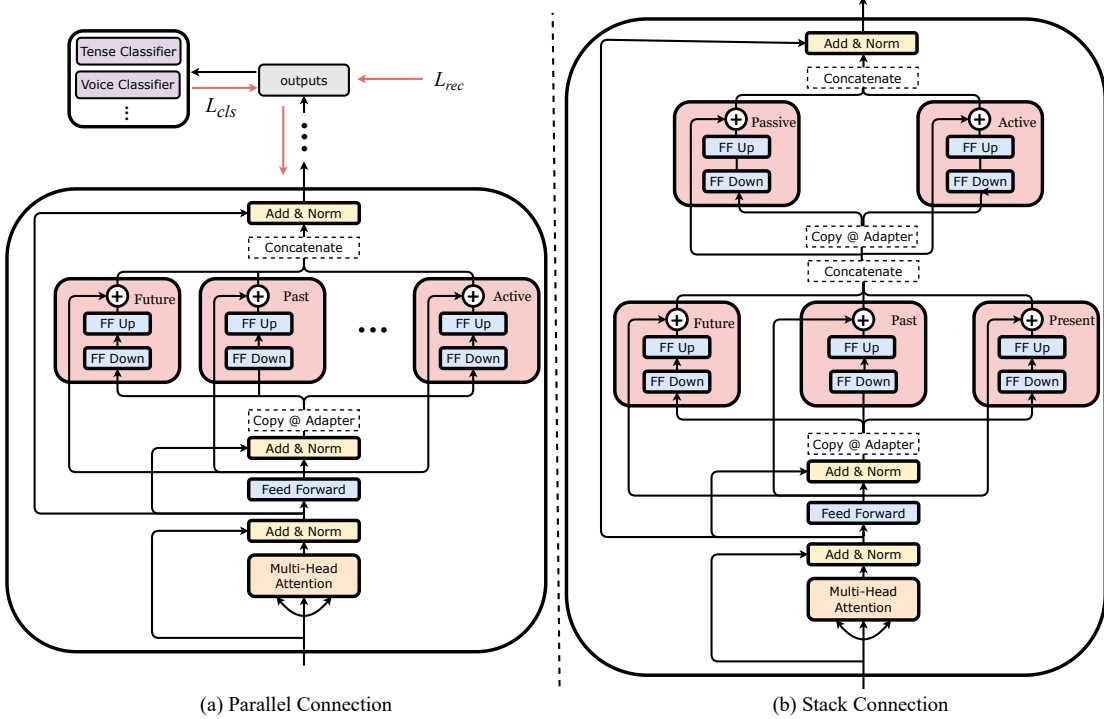

Figure 3: Adapter-TST Configurations - Left: Paralleling the adapters enables a single PLM to model different attributes simultaneously and generate multiple outputs in the corresponding target style. Right: Stacking the adapters for compositional editing in terms of different target styles at the same time. *Stack* connection is used for inference to verify the relevant attribute information captured by adapters.

for each attribute-specific adapter is about 3.17M, which is only 0.78% of the original BART-large model and 0.41% of the T5-large model, making the training process parameter efficient. Note that the original parameters of BART-large or T5-large are frozen during multiple-attribute TST training, and only the parameters of adapters are trainable and initialized randomly.

## 3.2 Adapter-TST Configurations

Adapter-TST has two configurations, *parallel* and *stack*, which support two multiple-attribute TST task settings: *multiple stylistic attribute outputs* and *compositional editing*, respectively. To better understand the two Configurations of Adapter-TST, we take the multiple-attribute TST task with *tense* and *voice* attributes as an example. *Tense* has three attribute values (*Future*, *Past*, *Present*), while *Voice* has two attribute values (*Passive*, *Active*). Thus, we add five attribute-specific adapters *Adapter(Future, Past, Present, Passive, Active)* to the base model for all the possible attribute values, respectively. Each adapter is employed to learn to generate sentences with corresponding attribute values while preserving the semantic content of the inputs.

**Parallel Connection.** We define the multi-

ple stylistic attribute outputs as follows: given a sentence $x = \{x_1, ..., x_n\}$ with $n$ tokens and $y_{tense}, y_{voice}$ labels, the Adapter-TST model is required to generate multiple outputs with all possible other attribute values at the same time. For instance, as shown in Figure 1(b), given an input sentence with present tense and active voice, the multiple-attribute TST models need to generate three sentences in the past tense, future tense, and passive voice simultaneously. The multiple stylistic attribute output setting requires TST models to capture all the stylistic attributes and have the capability of performing style transfer among the attribute values. Adapter-TST performs the multiple stylistic attribute output by utilizing the *Parallel* connection configuration shown in Figure 3(a). Specifically, we plug the paralleled adapters *Parallel(Future, Past, Present, Passive, Active)* into each transformer layer of the base model. During training, each training sample passes all the attribute-specific adapters, but adapters will take different actions according to the attribute values of input sentences. The adapter learns to reconstruct the input sentence for training samples with the same attribute value as an adapter. Conversely, when training samples with different attribute val-

ues, the adapter learns to transfer the attribute of the input sentence while preserving the original content. The outputs of all the adapters are concatenated together to the next layer. The replication is only performed once in the first transformer layer. In the latter transformer layers, we distribute the hidden states to corresponding adapters to make sure that the input of an adapter in the current layer is the output of the adapter with the same attribute value in the preceding layer.

**Stack Connection.** Compositional editing requires TST models to change multiple attributes simultaneously while preserving the original content. For instance, as shown in Figure 1(c), given an input sentence with present tense and active voice, the multiple-attribute TST models need to generate one sentence both in future tense and passive voice. Adapter-TST performs compositional editing by using the *Stack* connection method shown in Figure 3(b), where adapters belonging to the same attribute are parallelized because a sentence should only contain one attribute value for a specific attribute. Specifically, we have *Parallel(Future, Past, Present)* and *Parallel(Passive, Active)* for tense and voice attributes. The two sets of paralleled adapters are stacked as *Stack(Parallel(Future, Past, Present), Parallel(Passive, Active))* to learn to transfer multiple attributes. Similar to the *Parallel* connection method, the hidden states are replicated according to the number of adapters in the *Parallel* connection module. It's worth noting that, to demonstrate the attribute-specific adapters captured the attribute information, we only use the *Stack* connection method in inference time. During inference, we reload the parameters of adapters trained in multiple stylistic attribute outputs tasks and change the connection among the adapters to *Stack*.

### 3.3 Training Objectives.

The TST task aims to transfer the style of inputs while preserving the original semantic content. Thus, we train Adapter-TST with classification loss $L_{cls}$ for style transfer and reconstruction $L_{rec}$ for content preservation. During training, the original parameters of BART-large or T5-large are frozen, and only the parameters of adapters are trainable.

**Classification Loss** $L_{cls}$: The classification loss ensures that the transferred sentence conforms to the target attribute value. To this end, we first pre-train a TextCNN-based (Kim, 2014) binary at-

tribute classifier $D$ for each attribute, then apply the pre-trained attribute classifiers to guide the updates of adapters' parameters such that the output sentence is predicted to be in the target style:

$$L_{cls} = -\mathbb{E}_{(x,y)\sim D}[logP(y_t|x')] \qquad (1)$$

where $x'$ is sampled from the distribution of model outputs at each decoding time step, and $y_t$ is the target attribute value. Policy gradient algorithm (Sutton et al., 1999) is used to for discrete training with the attribute classifiers.

**Reconstruction Loss** $L_{rec}$: The reconstruction loss attempts to preserve the original content information in the transferred sentences. Specifically, the loss function constricts the adapters to capture informative features to reconstruct the original sentence using the learned representations. Formally, we define $L_{rec}$ as follows:

$$L_{rec} = -logP(x|z_i, y_i) \qquad (2)$$

where $y_i$ is the $i$-th attribute value of the input sentence, $z_i$ denotes the hidden representation extracted by the corresponding adapter. The input sentences are only reconstructed by the corresponding adapter and transferred by other adapters.

**Putting them together,** the final joint training loss $L$ is as follows:

$$L = (1 - \lambda)L_{rec} + \lambda L_{cls} \qquad (3)$$

Where $\lambda$ is a balancing hyper-parameter to ensure that the transferred sentence has the target style while preserving the original content.

## 4 Experiments

### 4.1 Experiment Setting

**Datasets.** We evaluate the proposed Adapter-TST model on sentiment transfer and multiple-attribute TST tasks using the Yelp[1] and StylePTB (Lyu et al., 2021) datasets, respectively. We adopt the train, development, and test split for the Yelp dataset as (Luo et al., 2019). Lyu et al. (2021) introduce StylePTB [2], a large-scale benchmark with compositions of multiple-attribute TST tasks which allow the modeling of fine-grained stylistic changes. In our experiments, we choose four subsets for multiple-attribute TST: Tense-Voice, Tense-PP-Front↔Back, Tense–PP-Removal, and Tense-ADJADV-Removal. Specifically, the four subsets

---

[1]https://github.com/luofuli/DualRL
[2]https://github.com/lvyiwei1/StylePTB

| Dataset | Train | Dev | Test |
|---|---|---|---|
| Yelp | 443K | 1,000 | 1,000 |
| Tense-voice | 28K | 1,538 | 1,564 |
| Tense-PP-Front↔Back | 5K | 270 | 284 |
| Tense-PP-Removal | 32K | 1,796 | 1,834 |
| Tense-ADJADV-Removal | 33K | 1,838 | 1,819 |

Table 1: Dataset statistics for Yelp and StylePTB.

include five attributes, tense with three attribute values (*Future*, *Past*, *Present*), voice with two attribute values (*Passive*, *Active*), proposition position with two attribute values (*Front*, *Back*), proposition removal with two attribute values (*Adding*, *Removing*), and adjectives&adverbs removal with two attribute values (*Adding*, *Removing*). Table 1 shows the training, validation, and test splits of the Yelp and StylePTB datasets used in our experiments.

**Baselines.** For sentiment transfer, we benchmark Adapter-TST against nine state-of-the-art TST models: *BackTrans* (Prabhumoye et al., 2018), *CrossAlign* (Shen et al., 2017), *DualRL* (Luo et al., 2019), *Unpaired* (Li et al., 2019), *UnsuperMT* (Zhang et al., 2018), *Style Transformer* (Dai et al., 2019), *DeleteOnly*, *Template*, and *Del&Retri* (Li et al., 2018). For multiple stylistic attribute outputs task, *Style Transformer* (Dai et al., 2019), a transformer-based model for single-attribute TST, is selected as a baseline. We train multiple *Style Transformer* models for each attribute and perform style transfer separately. For compositional editing, we use the trained *Style Transformer* models to perform sequential editing, which transfers one attribute after another to compare results with our model. We term this baseline as *Sequential Style Transformer* setup.

**Training.** The experiments were performed on an Ubuntu 20.04.3 LTS system with 24 cores, 128 GB RAM, and Nvidia RTX 3090. The model implementation is based on AdapterHub (Pfeiffer et al., 2020) and Huggingface Transformers (Wolf et al., 2020). For the balancing hyper-parameter $\lambda$, we choose the best-performed one from (0.9, 1) as the BART-large and T5-large models can copy the input without training with TST objectives.

### 4.2 Automatic Evaluation

We evaluate the proposed model and baselines on three criteria commonly used in TST studies: *transfer strength*, *content preservation*, and *fluency*. An attribute classifier is first pre-trained to predict the attribute label of the input sentence. The classifier is subsequently used to approximate the style trans-

| Model | ACC | BS | PPL | G |
|---|---|---|---|---|
| BackTrans | 94.5 | 0.88 | 11.3 | 1.95 |
| CrossAlign | 74.3 | 0.89 | 35.3 | 1.23 |
| DeleteOnly | 87.6 | 0.91 | 36.4 | 1.30 |
| Del&Retri | 90.2 | 0.91 | 34.0 | 1.34 |
| DualRL | 88.9 | **0.95** | 27.1 | 1.46 |
| Template | 83.7 | 0.92 | 47.2 | 1.18 |
| Unpaired | 50.6 | 0.91 | 53.1 | 0.95 |
| UnsuperMT | 96.2 | 0.93 | 33.5 | 1.39 |
| Style Transformer | 85.8 | **0.95** | 10.1 | 2.00 |
| Adapter-TST-BART | 90.1 | 0.91 | 8.2 | 2.15 |
| Adapter-TST-T5 | **97.3** | 0.89 | **4.8** | **2.62** |

Table 2: Performance of models on Yelp dataset (Sentiment Transfer Task). The best performances are **bold**.

fer accuracy (ACC) of the sentences' transferred attributes by considering the target attribute value as the ground truth. To quantitatively measure the amount of original content preserved after style transfer operations, we employ BERTscore (Zhang et al., 2019a) between style-transferred and original sentences. For fluency, We use GPT-2 (Radford et al., 2019) to measure the perplexity (PPL) of transferred sentences. The sentences with smaller PPL scores are considered more fluent. Finally, we compute the geometric mean of ACC, BERTscore, and 1/PPL. We take the inverse of the calculated perplexity score because a smaller PPL score corresponds to better fluency. When there is more than one accuracy in the multiple-attribute TST tasks, we use the average accuracy to compute G-score.

### 4.3 Automatic Evaluation Results

Table 2 shows the performance of the Adapter-TST model and the baselines on the sentiment transfer task. Adapter-TST has achieved the best G-score, outperforming the baselines. We observe that Adapter-TST achieves comparable performance on transfer strength and content preservation with 97.3% transfer accuracy and 0.89 BERTscore by only updating the parameters of adapters. With the impressive generative ability of the pre-trained BART-large and T5-large models, the Adapter-TST model can generate high-quality text in terms of fluency and completeness. The experiment results demonstrate Adapter-TST's ability to perform TST well and efficiently with fewer training parameters.

Table 3 presents the results of the proposed Adapter-TST model and Style Transformer baselines for the multiple stylistic attribute output task. Our Adapter-TST model achieves the highest G-score across all four datasets by simultaneously modeling multiple attributes using different

| Model | Attri | Tense-Voice | | | | | Tense-ADJADV-Removal | | | | |
|---|---|---|---|---|---|---|---|---|---|---|---|
| | | Tense | Voice | BS | PPL | G | Tense | Removal | BS | PPL | G |
| Style Transformer | single | 91.1 | - | 0.91 | 15.3 | 1.76 | 92.6 | - | 0.92 | 27.0 | 1.47 |
| Style Transformer | single | - | **87.2** | 0.85 | 11 | 1.89 | - | 83.7 | 0.93 | 21.7 | 1.53 |
| Adapter-TST-BART | multi | **96.9** | 81.9 | **0.96** | 4.7 | 2.63 | **96.2** | 76.5 | **0.95** | 11.8 | 1.91 |
| Adapter-TST-T5 | multi | 95.9 | 83.4 | 0.94 | **2.6** | **3.19** | 95.7 | **85.2** | 0.91 | **3.8** | **2.79** |
| Model | Attri | Tense-PP-Front↔Back | | | | | Tense-PP-Removal | | | | |
| | | Tense | F↔Back | BS | PPL | G | Tense | Removal | BS | PPL | G |
| Style Transformer | single | 95.7 | - | 0.83 | 6.8 | 2.27 | 94.9 | - | 0.91 | 27 | 1.47 |
| Style Transformer | single | - | **57.2** | 0.83 | 10.4 | 1.66 | - | 87.2 | 0.91 | 26.1 | 1.45 |
| Adapter-TST-BART | multi | 88.2 | 48.9 | **0.96** | 4 | 2.54 | **96** | 74.5 | **0.96** | 12.5 | **1.87** |
| Adapter-TST-T5 | multi | 75.9 | 48.2 | 0.83 | **1.7** | **3.12** | 95.6 | 95.0 | 0.92 | **3.8** | **2.85** |

Table 3: Automatic evaluation results of models on multiple stylistic attribute outputs task. The best performances are **bold**.

| Model | Tense-Voice | | | | | Tense-ADJADV-Removal | | | | |
|---|---|---|---|---|---|---|---|---|---|---|
| | Tense | Voice | BS | PPL | G | Tense | Removal | BS | PPL | G |
| Sequential Style Transformer | 80.2 | **88.1** | 0.85 | 22.2 | 1.48 | 88.6 | 90.0 | **0.89** | 42.2 | 1.23 |
| Adapter-TST-BART | **88.2** | 85.4 | 0.90 | 8.0 | 2.14 | **88.9** | **92.7** | 0.86 | 22 | 1.53 |
| Adapter-TST-T5 | 75.6 | 73.1 | **0.91** | 6.5 | 2.18 | 85.4 | 77.1 | 0.87 | **12.1** | **1.80** |
| Model | Tense-PP-Front↔Back | | | | | Tense-PP-Removal | | | | |
| | Tense | F↔Back | BS | PPL | G | Tense | Removal | BS | PPL | G |
| Sequential Style Transformer | 76.1 | **65.7** | 0.82 | 8.1 | 1.93 | **91.2** | 85.7 | 0.88 | 51.4 | 1.15 |
| Adapter-TST-BART | **88.2** | 50.0 | **0.92** | 4.9 | 2.35 | 90.1 | **88.2** | 0.86 | 20.9 | 1.54 |
| Adapter-TST-T5 | 73.5 | 51.4 | 0.88 | **1.8** | **3.13** | 82.1 | 81.7 | **0.91** | 19.7 | **1.56** |

Table 4: Automatic evaluation results of models on compositional editing task. The best performances are **bold**.

adapters. Adapter-TST performs well in transferring tense attributes, surpassing the baselines on three datasets. However, modeling multiple attributes together proves to be a more challenging task. While Adapter-TST exhibits a slight performance gap compared to the Style Transformer model in terms of transfer accuracy, it excels in generating fluent and coherent sentences while preserving the original content. This advantage allows Adapter-TST to outperform the baselines in content preservation and fluency. It is also worth noting that training multiple Style Transformers for the multiple-attribute TST tasks is computationally inefficient and expensive, unlike Adapter-TST.

To demonstrate that the attribute-specific adapters capture the corresponding attribute information, we evaluate the proposed Adapter-TST model on the compositional editing task. Note that the parameters of adapters trained in the multiple stylistic attribute outputs task are reloaded, and the connection method is changed to *Stack* for compositional editing. Table 4 shows the performance of the Adapter-TST and Sequential Style Transformer on the compositional editing task. The Adapter-TST model achieves the highest G-score across four datasets, similar to the results obtained in the multiple stylistic attribute output task. We observe that the average G-score of the multiple

stylistic attribute outputs task is 2.24, significantly higher than compositional editing's average G-score of 1.89. The difference in the average G-score highlights the challenge of the compositional editing task. Interestingly, Adapter-TST achieves comparable performance on style transfer accuracy over attributes, indicating that the attribute-specific adapters effectively capture the stylistic attributes.

### 4.4 Human Evaluation

We conducted a human-based evaluation study to assess the performance of the Adapter-TST model in handling multiple-attribute TST tasks. The study involved randomly sampling 200 sentences from the Tense-Voice dataset. Both Adapter-TST and the baselines were used to generate multiple stylistic attribute outputs and perform compositional editing on the sampled sentences. Two linguistic researchers evaluated the generated sentences based on three criteria used in automated evaluation. To measure *transfer strength*, evaluators indicated whether the sentences were in the target attribute value (e.g., future tense, passive voice) using a true/false indicator. For *content preservation*, evaluators rated the amount of preserved content on a 5-point Likert scale, ranging from no content preserved (1) to all content preserved (5). Fluency was assessed on a 5-point Likert scale, where 1 represented unreadable sentences with numerous

| Model | Attribute | Tense-Voice | | | | |
|---|---|---|---|---|---|---|
| | | Tense | Voice | BS | PPL | G |
| Style Transformer | Tense | 90.0 | - | 3.72 | 3.23 | 10.26 |
| Style Transformer | Voice | - | **74.0** | 2.16 | 2.42 | 7.29 |
| Adapter-TST (ours) | Tense+Voice | **99** | 67.0 | **3.74** | **3.58** | **10.35** |
| Sequential Style Transformer | Tense+Voice | 81.0 | **85.0** | 2.56 | 2.88 | 8.49 |
| Adapter-TST (ours) | Tense+Voice | **93.0** | 82.0 | **3.19** | **3.00** | **9.43** |

Table 5: Human evaluation results of models on both multiple stylistic attribute outputs and compositional editing tasks. The best performances are **bold**.

| Target Style | Source Sentence | Style Transformer | Adapter-TST |
|---|---|---|---|
| Future | The plan lacked a withdrawal timetable. | The plan will lack be had by the buy-out group. | The plan will have a withdrawal timetable. |
| Past | Some issues will be helped by higher earnings. | Some issues were helped by higher earnings by some issues. | Some issues were helped by higher earnings. |
| Present | And he will question the white house dedication. | And he question the white house and he | And he says the white house dedication. |
| Future+passive | Litigation sciences doesn't make moral distinctions. | Litigation transportation will not make fuel had and this teaches us and no she will be had by either. | Moral distinctions will be done by Litigation sciences. |
| Past+Active | Third high yields are offered by them. | Third nutmeg yields listed night board them third period earnings stage | Third high yields offered to them. |

Table 6: Qualitative results for transfer to different target style combination across different models. Different colors highlight the transferred segments contributing to the target style.

grammatical errors, and 5 indicated perfect and fluent sentences. To reduce biases, the model names were concealed, and the order of the models was randomized when displaying the generated sentences. This ensured that evaluators were unaware of which model generated the sentences they were evaluating.

## 4.5 Human Evaluation Results

Table 5 shows the evaluation results. The style transfer accuracy of the models was computed using the binary feedback from the evaluators. The average scores for the criteria of content preservation and fluency were calculated using the 5-point Likert scores. Adapter-TST is observed to outperform the baselines in content preservation, fluency, and G-score. Adapter-TST is also rated to generate more syntactically sound and fluent sentences compared to the baselines. We can also observe that there is still a style transfer accuracy drop of Adapter-TST on attribute Voice when modeling multiple attributes at the same time. These results align with the automatic evaluations and demonstrate Adapter-TST's effectiveness in performing multiple-attribute TST well and efficiently.

## 5 Case Study

We conducted case studies to showcase the style transferred outputs of both the Adapter-TST and Style Transformer models. Randomly sampled examples and their corresponding outputs are presented in Table 6, specifically for the Tense-Voice dataset. Our findings reveal that Adapter-TST successfully transfers the style while preserving the content and sentence structure in multiple-attribute TST tasks. In contrast, the Style Transformer model generates sentences with grammatical errors, making it challenging to determine if the style transfer was successful. Moreover, the Style Transformer model performs poorly in the task of compositional editing due to its inherent complexity. Despite the difficulty of compositional editing, Adapter-TST is capable of generating fluent sentences that preserve the original content.

## 6 Conclusion

In this paper, we introduced a parameter-efficient framework, Adapter-TST with different neural adapters to capture different attribute information for multiple-attribute TST tasks. During training, the original parameters of BART-large were frozen, and only the adapters' parameters were optimized to relax the dependence on computational resources and supervised data. We conducted extensive ex-

periments on traditional sentiment transfer and multiple-attribute TST tasks. The automatic and human-based evaluation results showed that the attribute-specific adapters in Adapter-TST is able to capture relevant stylistic attributes to transfer the style while preserving the original content successfully. Our case studies also demonstrated that Adapter-TST was able to generate high-quality text in the target style. For future work, we will continue to improve TST models' ability to model multiple attributes in terms of quality and efficiency. We will also explore plugging Adapter-TST on other PLMs and evaluate its effectiveness.

## 7 Limitations

This work has two limitations. First, there is a style transfer accuracy reduction on one of the attributes, while the proposed model models multiple attributes simultaneously. Explorations on improving TST models' ability to handle multiple-attribute TST tasks and the dependency among attributes are potential directions in this field. Second, even though we have frozen the parameters of the pre-trained BART-large model to improve parameter efficiency, we still need to run BART-large model to extract representations for performing TST tasks.

## 8 Ethics Statement

The ethical implications of using large language models trained on data containing unchecked biases are acknowledged. As with any generative task, style transfer also has the potential for misuse, such as fact distortion, plagiarism, etc. The paper aims to demonstrate the academic utility of the proposed framework. This solution must be paired with strict checks for misrepresentation, offensiveness, and bias to adhere to ethical standards.

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
