# OpenReview forum: "Adapter-TST: A Parameter Efficient Method for Multiple-Attribute Text Style Transfer"
_EMNLP/2023/Conference — EMNLP 2023 Findings_

### Official Review · Reviewer_f2rF · 2023-08-03

**Soundness:** 4

**Excitement:**

3: Ambivalent: It has merits (e.g., it reports state-of-the-art results, the idea is nice), but there are key weaknesses (e.g., it describes incremental work), and it can significantly benefit from another round of revision. However, I won't object to accepting it if my co-reviewers champion it.

**Missing References:**

Controllable Unsupervised Text Attribute Transfer via Editing Entangled Latent Representation.

**Paper Topic And Main Contributions:**

The paper proposes an adapter-based multi-attribute text style transfer model with the parallel and stacked connection configurations. The authors focusing on an issue of the current PLM fine-tuning where the training examples are limited and the PLM contains a huge amount of parameters which may lead to overfitting.

**Reasons To Accept:**

The paper reads very smoothly and is an enjoyable read. The model is explained clearly and the paper is well-structured.

The method proposed is a new application scenario extension of the Adapter framework.

**Reasons To Reject:**

The experiments only utilized one backbone model, potentially leading to an evaluation bias concerning the model's effectiveness. To address this concern, the author should investigate and assess the impact of employing various backbones on the performance of the adapters.

The human evaluation lacks specific quality details, such as the number of examples used for evaluation and the number of workers hired to conduct the evaluations. Including these details is essential to ensure transparency and replicability of the evaluation process.

The comparison with prior works is insufficient, raising concerns about the effectiveness of the proposed model.

Missing some related references, e.g., FGIM, Wang et al., 2019.

**Reproducibility:**

4: Could mostly reproduce the results, but there may be some variation because of sample variance or minor variations in their interpretation of the protocol or method.

**Reviewer Confidence:**

4: Quite sure. I tried to check the important points carefully. It's unlikely, though conceivable, that I missed something that should affect my ratings.

---

> ### Author Rebuttal · Authors · 2023-08-29
>
> Thank you for the insightful feedback. We hope that we have comprehensively addressed the concerns and questions you presented. Based on the valuable discussion, we will make the necessary updates to our paper.
>
> ---
>
> ***W1**: The experiments only utilized one backbone model, potentially leading to an evaluation bias concerning the model's effectiveness. To address this concern, the author should investigate and assess the impact of employing various backbones on the performance of the adapters.*
>
> **[Response to W1]**
>
> We value your feedback on improving the evaluation of our AdapterTST’s generalizibility to other backbones. Our framework is designed to be compatible with various models, highlighting its flexibility. While we initially opted for BART-Large due to its known effectiveness in similar tasks, we agree that tests with models such as T5 and LLaMA would further demonstrate AdapterTST's versatility. Due to computational constraints and time limitations, we could only include the results for applying AdapterTST on T5. Below table shows the benchmarking results on Yelp dataset.
>
> |Model| ACC| Bertscore| PPL| G-Score|
> |-|-|-|-|-|
> |BackTrans |94.5 |0.88 |11.3 |1.95|
> |CrossAlign |74.3| 0.89| 35.3| 1.23|
> |DeleteOnly |87.6 |0.91 |36.4 |1.30|
> |Del&Retri |90.2| 0.91| 34.0| 1.34|
> |DualRL |88.9| 0.95| 27.1 |1.46|
> |Template |83.7| 0.92| 47.2| 1.18|
> |Unpaired |50.6| 0.91| 53.1| 0.95|
> |UnsuperMT |96.2| 0.93| 33.5| 1.39|
> |Style Transformer |85.8 |0.95 |10.1 |2.00|
> |Adapter-TST_BART-Large| 90.1| 0.91| 8.2| 2.15|
> |Adapter-TST_T5-Large| 97.3|0.89|4.8|2.62 |
>
> We noted that applying Adapter-TST_T5-Large yield the best performance on the Yelp dataset. We will complete the experiments and apply Adapter-TST on other LLMs such as LLaMA, and test the models on multiple stylistic attribute dataset. The updated results will be reported in the camera-ready version of our paper.
>
> ---
>
> ***W2**: The human evaluation lacks specific quality details, such as the number of examples used for evaluation and the number of workers hired to conduct the evaluations. Including these details is essential to ensure transparency and replicability of the evaluation process.*
>
> **[Response to W2]**
>
> You're absolutely right in emphasizing the importance of detailed transparency in the human evaluation process. We apologize for any oversight. To clarify, we have included these specific details between Lines 500-508. The study involved a random sampling of 200 sentences from the Tense-Voice dataset. For each of these sentences, both Adapter-TST and the baselines were utilized to generate outputs based on multiple stylistic attributes and to execute compositional editing. Two linguistic researchers were then engaged to assess these generated sentences using three specific criteria, similar to the ones utilized in automated evaluations. We hope this provides clearer insights into our evaluation process.
>
> ---
>
> ***W3**: The comparison with prior works is insufficient, raising concerns about the effectiveness of the proposed model.*
>
> **[Response to W3]**
>
> Thank you for pointing out the importance of using stronger baselines for our Yelp dataset evaluation. We are dedicated to refining our experimental design to maintain rigor. To address this, we're incorporating newer, state-of-the-art baselines relevant to the Yelp dataset. This ensures a more robust comparison of Adapter-TST's performance.
>
> We've included the recent baselines [1] and [2] for the Yelp dataset and assessed their performance using the authors' original outputs. The table below details the results of Adapter-TST alongside these added baselines. Notably, comparable results for PromptAndRerank are documented in [3], based on human evaluations.
>
> |Model|Data|Acc|BERTScore|Fluency (PPL)| G-score |
> |-|-|-|-|-|-|
> | Adapter-TST| Yelp | 90.1 |    0.91   |   8.2   |   2.15  |
> | CRF [2]| Yelp | 92.0 |    0.96   |   26.8  |   1.49  |
> | PromptAndRerank 0-shot [1,3] | Yelp | 53.0 |    0.94   |   20.5  |   1.34  |
> | PromptAndRerank 4-shot [1,3] | Yelp | 63.6 |    0.95   |   20.6  |   1.43  |
>
> - [1] Prompt-and-Rerank: A Method for Zero-Shot and Few-Shot Arbitrary Textual Style Transfer with Small Language Models (Suzgun et al., EMNLP 2022)
> - [2] Tagging Without Rewriting: A Probabilistic Model for Unpaired Sentiment and Style Transfer (Shuo, WASSA 2022)
> - [3] Fluency Matters! Controllable Style Transfer with Syntax Guidance (Han & Sohn, WASSA 2023)
>
> ---
>
> ***Others**: Missing References*
>
> We will add the suggested reference and perform a more thorough literature review in the updated version of our paper.

---

### Official Review · Reviewer_Jb8X · 2023-08-04

**Typos Grammar Style And Presentation Improvements:** N/A
**Soundness:** 3

**Excitement:**

2: Mediocre: This paper makes marginal contributions (vs non-contemporaneous work), so I would rather not see it in the conference.

**Missing References:**

See Reasons To Reject

**Paper Topic And Main Contributions:**

The paper introduces an adapter-based approach for the multiple-attribute text style transfer task.  In short, the processed method Adapter-TST utilizes a series of adapters to model different types of attribute information. As a parameter-efficient method, Adapter-TST achieves better performance with a very low number of training parameters compared to the previous method.

**Questions For The Authors:**

See Reasons To Reject

**Reasons To Accept:**

1. The paper was well-written and easy to follow.
2. How to use parametric efficient methods in style transfer tasks is an important field worth exploring.

**Reasons To Reject:**

1. The contribution of the whole paper is limited.  Using adapter-based PLMs is a common strategy for text generation tasks, even those involving multi-attribute-based generation tasks, such as [2][3]. At the same time, the adapter used in this paper is not significantly different from the previous common methods, nor is it optimized for tasks.
2. This paper designed the method only with BART as the backbone. For this reason, I think it might be difficult to call this particular approach a "parameter-efficient framework" because its generality has yet to be proven.
3. There are serious deficiencies and possible unfair comparisons in the experimental part. The main reasons are as follows:
3.1 Baselines selected in the experiment are seriously missing. The experiment only compares the proposed method with the non-parameter-efficient method Styletransformer, while other parameter-efficient methods are ignored, especially prompt-learning-based methods, such as[1][2][3]. Although these methods are not applied to style transfer, it is clear that they are general-purpose and can be easily migrated to this task, and [2][3] also perform a similar multi-attribute text generation task while using adapter-based PLMs as baselines.
3.2 Unfair comparison with baselines.  The backbone of Adapter-TST is BART-Large while the backbone of the baseline is Transformer,  it is difficult to determine whether the performance gains are due to the use of PLMs or the proposed approach.

References:

[1] Prefix-Tuning: Optimizing Continuous Prompts for Generation.

[2] Controllable Natural Language Generation with Contrastive Prefixes.

[3] Tailor: A Soft-Prompt-Based Approach to Attribute-Based Controlled Text Generation.

**Reproducibility:**

4: Could mostly reproduce the results, but there may be some variation because of sample variance or minor variations in their interpretation of the protocol or method.

**Reviewer Confidence:**

5: Positive that my evaluation is correct. I read the paper very carefully and I am very familiar with related work.

---

> ### Author Rebuttal · Authors · 2023-08-29
>
> Thank you for the feedback and questions. We hope that we have comprehensively addressed the concerns and questions you presented. Based on the valuable discussion, we will make the necessary updates to our paper.
>
> ---
>
> ***W1**: The contribution of the whole paper is limited. Using adapter-based PLMs is a common strategy for text generation tasks, even those involving multi-attribute-based generation tasks, such as [2][3]. At the same time, the adapter used in this paper is not significantly different from the previous common methods, nor is it optimized for tasks.*
>
> **[Response to W1]**
>
> While we concur that adapter-based PLMs have seen their utilization in text generation tasks, it's pertinent to delineate the distinctions and innovations brought forth by our Adapter-TST method. Indeed, the cited works [2] and [3] have employed prompt learning-based techniques, and while they share some methodological similarities with our approach, the application and end goal differ considerably.
>
> **Our primary contribution stems from the specialized application of adapters to manage multi-attribute text style transfer, a domain that, to the best of our knowledge, hasn't been fully explored with this methodology.** The innate strength of our model rests in the two distinct connection methods tailored to cater to a variety of multi-attribute TST tasks.
>
> Furthermore, the nuance of our approach materializes in its parallel architecture, allowing different adapters to specialize in disparate attributes simultaneously within a single sentence. This not only promotes efficiency but also ensures that each attribute is treated with the required depth and precision.
>
> We understand that the advancement of any scientific field is predicated upon building upon prior knowledge, and while our approach derives inspiration from previous methods, it introduces nuances that we believe make a meaningful contribution to the area of multi-attribute text style transfer.
>
> ---
>
> ***W2**: This paper designed the method only with BART as the backbone. For this reason, I think it might be difficult to call this particular approach a "parameter-efficient framework" because its generality has yet to be proven.*
>
> **[Response to W2]**
>
> Our framework is designed to be compatible with various models, highlighting its flexibility. While we initially opted for BART-Large due to its known effectiveness in similar tasks, we agree that tests with models such as T5 and LLaMA would further demonstrate AdapterTST's versatility. We sincerely regret that due to computational constraints and time limitations, we could only include the results for applying AdapterTST on T5. Below table shows the benchmarking results on Yelp dataset.
>
> |Model| ACC| Bertscore| PPL| G-Score|
> |-|-|-|-|-|
> |BackTrans |94.5 |0.88 |11.3 |1.95|
> |CrossAlign |74.3| 0.89| 35.3| 1.23|
> |DeleteOnly |87.6 |0.91 |36.4 |1.30|
> |Del&Retri |90.2| 0.91| 34.0| 1.34|
> |DualRL |88.9| 0.95| 27.1 |1.46|
> |Template |83.7| 0.92| 47.2| 1.18|
> |Unpaired |50.6| 0.91| 53.1| 0.95|
> |UnsuperMT |96.2| 0.93| 33.5| 1.39|
> |Style Transformer |85.8 |0.95 |10.1 |2.00|
> |Adapter-TST_BART-Large| 90.1| 0.91| 8.2| 2.15|
> |Adapter-TST_T5-Large| 97.3|0.89|4.8|2.62 |
>
> We noted that applying Adapter-TST_T5-Large yield the best performance on the Yelp dataset. We will complete the experiments and apply Adapter-TST on other LLMs such as LLaMA, and test the models on multiple stylistic attribute dataset. The updated results will be reported in the camera-ready version of our paper.
>
> ---
>
> ***W3**: There are serious deficiencies and possible unfair comparisons in the experimental part. The main reasons are as follows: 3.1 Baselines selected in the experiment are seriously missing. The experiment only compares the proposed method with the non-parameter-efficient method Styletransformer, while other parameter-efficient methods are ignored, especially prompt-learning-based methods, such as[1][2][3]. Although these methods are not applied to style transfer, it is clear that they are general-purpose and can be easily migrated to this task, and [2][3] also perform a similar multi-attribute text generation task while using adapter-based PLMs as baselines. 3.2 Unfair comparison with baselines. The backbone of Adapter-TST is BART-Large while the backbone of the baseline is Transformer, it is difficult to determine whether the performance gains are due to the use of PLMs or the proposed approach.
>
> References:
> - [1] Prefix-Tuning: Optimizing Continuous Prompts for Generation.
> - [2] Controllable Natural Language Generation with Contrastive Prefixes.
> - [3] Tailor: A Soft-Prompt-Based Approach to Attribute-Based Controlled Text Generation.*
>
> **[Response to W3]**
>
> Your concerns regarding the experimental section of our paper are noted. Firstly, we'd like to address the misconception regarding the works [2] and [3]. The objective of these two studies is controllable text generation, in which the models are trained using language modeling objectives with original text serving as the ground truth. During evaluation, these models are instructed to complete input sentences while controlling for specific attributes. Our Adapter-TST, on the other hand, is tailored for multi-attribute TST tasks where the primary goal is to edit multiple attributes of the input text in the absence of parallel data. Essentially, our model carries out sequence-to-sequence generation tasks without the aid of parallel data. This distinction necessitates the employment of reconstruction loss for content retention and classification loss for attribute guidance. Therefore, our approach substantially deviates from the methods presented in [2] and [3], making their seamless migration to our task challenging.
>
> To provide more clarity on the two data settings in TST:
>
> - Parallel Setting: Under this framework, TST models are trained utilizing known pairs of text, each having distinct styles.
> -Non-parallel Setting: Here, TST models aim to modify the style of text without the benefit of knowing the matching text pairs in alternate styles.
>
> For a more in-depth understanding of the differences between our proposed methodology and the cited works, we encourage a thorough reading of the following survey papers on TST:
> - [a] Deep Learning for Text Style Transfer: A Survey
> - [b] Text Style Transfer: A Review and Experimental Evaluation.

---

### Official Review · Reviewer_2k94 · 2023-08-05

**Soundness:** 4

**Excitement:**

3: Ambivalent: It has merits (e.g., it reports state-of-the-art results, the idea is nice), but there are key weaknesses (e.g., it describes incremental work), and it can significantly benefit from another round of revision. However, I won't object to accepting it if my co-reviewers champion it.

**Missing References:**

**TST Literature**
* Eric Malmi, Aliaksei Severyn, and Sascha Rothe. 2020. Unsupervised Text Style Transfer with Padded Masked Language Models. arXiv preprint arXiv:2010.01054.
* Emily Reif, Daphne Ippolito, Ann Yuan, Andy Coenen, Chris Callison-Burch, and Jason Wei. 2022. A Recipe for Arbitrary Text Style Transfer with Large Language Models. In Proceedings of the 60th Annual Meeting of the Association for Computational Linguistics (Volume 2: Short Papers), pages 837848, Dublin, Ireland. Association for Computational Linguistics.
* Mirac Suzgun, Luke Melas-Kyriazi, and Dan Jurafsky. 2022. Prompt-and-Rerank: A Method for Zero-Shot and Few-Shot Arbitrary Textual Style Transfer with Small Language Models. In Proceedings of the 2022 Conference on Empirical Methods in Natural Language Processing, pages 2195–2222, Abu Dhabi, United Arab Emirates. Association for Computational Linguistics.
* Akhilesh Sudhakar, Bhargav Upadhyay, and Arjun Maheswaran. 2019. "Transforming" Delete, Retrieve, Generate Approach for Controlled Text Style Transfer. In Proceedings of the 2019 Conference on Empirical Methods in Natural Language Processing and the 9th International Joint Conference on Natural Language Processing (EMNLP-IJCNLP), pages 3269–3279.
* Shamik Roy, Raphael Shu, Nikolaos Pappas, Elman Mansimov, Y. Zhang, Saab Mansour and Dan Roth. “Conversation Style Transfer using Few-Shot Learning.” ArXiv abs/2302.08362 (2023).
* Kalpesh Krishna, Deepak Nathani, Xavier Garcia, Bidisha Samanta, and Partha Talukdar. 2022. Few-shot Controllable Style Transfer for Low-Resource Multilingual Settings. In Proceedings of the 60th Annual Meeting of the Association for Computational Linguistics (Volume 1: Long Papers), pages 7439–7468, Dublin, Ireland. Association for Computational Linguistics.

**LLaMa-Adapter**
* Renrui Zhang, Jiaming Han, Aojun Zhou, Xiangfei Hu, Shilin Yan, Pan Lu, Hongsheng Li, Peng Gao and Yu Jiao Qiao. “LLaMA-Adapter: Efficient Fine-tuning of Language Models with Zero-init Attention.” ArXiv abs/2303.16199 (2023).

**Paper Topic And Main Contributions:**

This paper presents Adapter-TST, a straightforward yet effective and light-weight method for addressing the challenging task of multiple-attribute textual style transfer (TST). In contrast to single-attribute TST, which focuses on changing one stylistic property at a time, Adapter-TST deals with simultaneously altering multiple stylistic properties—such as sentiment, tense, voice, formality, and politeness. The Adapter-TST approach involves inserting simple neural adapters (consisting a feed-forward down-project layer, followed by a nonlinear activation function, followed by a feed forward up-project layer, along with a skip-connection layer between two projection layers) into a pre-trained network, like a pre-trained BART model, to capture and change diverse attribute information. The weights (parameters) of the original network are kept frozen during training, and only the additional adapter layers are trained. Adapter-TST offers two configurations, "parallel" and "stack," enabling models to perform compositional style transformation (text editing).

In order to validate the effectiveness of Adapter-TST, the authors employ BART-Large as their backbone model and conduct experiments on the Yelp and StylePTB datasets. Their results indicate that Adapter-TST often performs on par with, not outperforms, several baseline methods, including BackTrans, CrossAlign, DualRL, and StyleTransformer, along certain evaluation metrics. Although human evaluation shows positive outcomes for Adapter-TST, the improvements do not appear to be statistically significant.

**Questions For The Authors:**

* Question A. Could you please provide more details about the training of the adapter layers? For instance, how many epochs did you train your models? How important is the classification loss?
* Question B. How crucial are the parallel connections? Do you think that stack connections might be enough to perform compositional text editing?
* Question C. Have you considered and conducted experiments using other models, in addition to BART-Large?
* Question D. Would your proposed method demonstrate favorable performance if trained on a combination of multiple textual style transfer datasets, such as sentiment, formality, grammar correction, and others?

**Reasons To Accept:**

* Textual style transfer with multiple attributes remains a relatively unexplored and challenging task within the field of NLP. This paper focuses on this intriguing problem and introduces an intuitive, effective, and light-weight approach to tackle it.
* The paper demonstrates a clear motivation for studying the problem of multiple-attribute textual style transfer and effectively outlines their research objectives. Moreover, the contributions of their work are well-defined, and the paper is written in a coherent manner overall, making it easy to understand both the arguments and results presented.
* This work has the potential to be of interest to not only the TST community but also the broader NLG community, as it proposes a parameter-efficient method for compositional text editing. (That said, the paper lacks clarity regarding the amount of fine-tuning data required to achieve satisfactory task performance and the overall generalizability of the proposed Adapter-TST approach. Further investigation on these aspects would be rather beneficial to better understand the practical implications of this method.)
* On the whole, the experimental setup seems to be sound and thorough.

**Reasons To Reject:**

* While this limitation is not a significant reason for rejection, I believe that the authors could enhance the credibility of their proposed approach by demonstrating its generalizability and robustness. Currently, the focus is solely on one true multi-attribute TST dataset (StylePTB) and one simple model type (BART-Large). The inclusion of an additional dataset would strengthen their claims and offer a clearer demonstration of Adapter-TST’s efficacy. Moreover, a more comprehensive analysis of their findings is needed in my opinion, as the paper, in its current form, offers only a surface-level discussion of the results.
* This is a relatively minor concern, but I fear that the paper does not sufficiently make reference to the recent studies on textual style transfer and consider stronger baselines for the Yelp dataset.
* In my opinion, the authors missed an opportunity to include two other simple yet potentially strong baselines in their evaluation: an Alpaca (or LLaMa) model with and without adapters. Considering these instruction-tuned models could have provided valuable insights even under a zero-shot setting. Comparing their proposed approach against such baselines could have further enriched the analysis and strengthened the overall study.

**Reproducibility:**

4: Could mostly reproduce the results, but there may be some variation because of sample variance or minor variations in their interpretation of the protocol or method.

**Reviewer Confidence:**

5: Positive that my evaluation is correct. I read the paper very carefully and I am very familiar with related work.

**Typos Grammar Style And Presentation Improvements:**

* In general, the paper is well-written and easy to follow; however, there is room for improvement in the readability and flow of Sections 3.2, 4.2, and 4.3.
* L170: 2019).To leverage → 2019. To leverage // Space needed between the period and the word “To”
* In Table 2, the abbreviations "BS" and "G" are not explicitly defined, leaving readers uncertain about their meanings. It appears that "G" possibly represents the geometric mean of ACC, BERTscore, 434, and 1/PPL.
* However, the rationale behind choosing the geometric mean of these specific metrics remains unclear. It would be beneficial for the authors to provide further explanation and intuition behind this choice to help readers better understand the motivation and significance of using the geometric mean as a composite metric.

---

> ### Author Rebuttal · Authors · 2023-08-29
>
> Thank you for the insightful feedback and questions. Your thought-provoking questions have significantly contributed to enhancing our paper's quality. We hope that we have comprehensively addressed the concerns and questions you presented. Based on the valuable discussion, we will make the necessary updates to our paper.
>
> ---
>
> ***W1**: While this limitation is not a significant reason for rejection, I believe that the authors could enhance the credibility of their proposed approach by demonstrating its generalizability and robustness. Currently, the focus is solely on one true multi-attribute TST dataset (StylePTB) and one simple model type (BART-Large). The inclusion of an additional dataset would strengthen their claims and offer a clearer demonstration of Adapter-TST’s efficacy. Moreover, a more comprehensive analysis of their findings is needed in my opinion, as the paper, in its current form, offers only a surface-level discussion of the results.*
>
> **[Response to W1]**
>
> **Comment on Dataset Limitation.** You've rightly emphasized the importance of diversifying our experimental datasets to validate the generalizability of AdapterTST. We understand that including another multi-attribute TST dataset would be ideal for demonstrating the adaptability of our approach. However, the field of multi-attribute TST is still nascent, and there is a scarcity of publicly available datasets tailored for such tasks. Our intention in focusing on the StylePTB dataset was to work with a reliable and comprehensive resource. That said, our hope is that the advancements we are introducing with AdapterTST will catalyze further research in this domain, leading to the creation of more datasets. We are actively monitoring for suitable datasets, and we will incorporate them into our experiments as they become available. We will add this discussion to the limitation section of our paper.
>
> **Comment on Model Evaluation.** Your point regarding the broader applicability of AdapterTST to models other than BART-Large is well-taken. Our framework is designed to be model-agnostic, emphasizing its versatility. While BART-Large was our initial choice given its prominence and efficiency in related tasks, we concur that experimenting with models like T5 and LLaMA will underscore AdapterTST's adaptability. We sincerely regret that due to computational constraints and time limitations, we could only include the results for applying AdapterTST on T5. Below table shows the benchmarking results on Yelp dataset.
>
> |Model| ACC| Bertscore| PPL| G-Score|
> |-|-|-|-|-|
> |BackTrans |94.5 |0.88 |11.3 |1.95|
> |CrossAlign |74.3| 0.89| 35.3| 1.23|
> |DeleteOnly |87.6 |0.91 |36.4 |1.30|
> |Del&Retri |90.2| 0.91| 34.0| 1.34|
> |DualRL |88.9| 0.95| 27.1 |1.46|
> |Template |83.7| 0.92| 47.2| 1.18|
> |Unpaired |50.6| 0.91| 53.1| 0.95|
> |UnsuperMT |96.2| 0.93| 33.5| 1.39|
> |Style Transformer |85.8 |0.95 |10.1 |2.00|
> |Adapter-TST_BART-Large| 90.1| 0.91| 8.2| 2.15|
> |Adapter-TST_T5-Large| 97.3|0.89|4.8|2.62 |
>
> We noted that applying Adapter-TST_T5-Large yield the best performance on the Yelp dataset. We will complete the experiments and apply Adapter-TST on other LLMs such as LLaMA, and test the models on multiple stylistic attribute dataset. The updated results will be reported in the camera-ready version of our paper.
>
> **In-depth Analysis.** We also recognize the need for a more detailed analysis of our results. In our eagerness to share the promise of AdapterTST with the community, we may have oversimplified certain aspects. We will strive to provide a deeper discussion, shedding light on the nuances and potential implications of our findings.
>
> ---
>
> ***W2**: This is a relatively minor concern, but I fear that the paper does not sufficiently make reference to the recent studies on textual style transfer and consider stronger baselines for the Yelp dataset.&
>
> **[Response to W2]**
>
> Thank you for pointing out the importance of using stronger baselines for our Yelp dataset evaluation. We are dedicated to refining our experimental design to maintain rigor. To address this, we're incorporating newer, state-of-the-art baselines relevant to the Yelp dataset. This ensures a more robust comparison for Adapter-TST's performance.
>
> We've included the recent baselines [1] and [2] for the Yelp dataset and assessed their performance using the authors' original outputs. The table below details the results of Adapter-TST alongside these added baselines. Notably, comparable results for PromptAndRerank are documented in [3], based on human evaluations.
>
> |Model|Data|Acc|BERTScore|Fluency (PPL)| G-score |
> |-|-|-|-|-|-|
> | Adapter-TST| Yelp | 90.1 |    0.91   |   8.2   |   2.15  |
> | CRF [2]| Yelp | 92.0 |    0.96   |   26.8  |   1.49  |
> | PromptAndRerank 0-shot [1,3] | Yelp | 53.0 |    0.94   |   20.5  |   1.34  |
> | PromptAndRerank 4-shot [1,3] | Yelp | 63.6 |    0.95   |   20.6  |   1.43  |
>
> - [1] Prompt-and-Rerank: A Method for Zero-Shot and Few-Shot Arbitrary Textual Style Transfer with Small Language Models (Suzgun et al., EMNLP 2022)
> - [2] Tagging Without Rewriting: A Probabilistic Model for Unpaired Sentiment and Style Transfer (Shuo, WASSA 2022)
> - [3] Fluency Matters! Controllable Style Transfer with Syntax Guidance (Han & Sohn, WASSA 2023)
>
> ---
>
> ***W3**: In my opinion, the authors missed an opportunity to include two other simple yet potentially strong baselines in their evaluation: an Alpaca (or LLaMa) model with and without adapters. Considering these instruction-tuned models could have provided valuable insights even under a zero-shot setting. Comparing their proposed approach against such baselines could have further enriched the analysis and strengthened the overall study.*
>
> **[Response to W3]**
>
> Thank you for pointing out the potential of including instruction-tuned models like Alpaca (or LLaMa) as baselines in our evaluation. We wholeheartedly agree with your assessment; such models can indeed offer additional dimensions to our study, especially in a zero-shot context. The reason we initially refrained from incorporating these models was primarily due to the intricacies of collecting instruction-tuning data for the multi-attribute TST domain. As this area of research is still emerging, there exists a dearth of readily available data tailored for such specific instruction-tuning tasks. We will certainly consider this in our future works.
>
> ---
>
> ***Q1**: Could you please provide more details about the training of the adapter layers? For instance, how many epochs did you train your models? How important is the classification loss?*
>
> **[Response to Q1]**
>
> We trained our Adapter-TST models for a total of 5 epochs. This was determined after several rounds of experimentation, ensuring that the model achieves robust performance without overfitting.
>
> Regarding the classification loss: its importance cannot be understated. The classification loss plays a pivotal role as it steers the learning of the adapter layers towards capturing the stylistic attributes effectively. Without the incorporation of the classification loss, we observed that the model tends to produce outputs that merely echo the input sentences, leading to an ineffective style transfer.
>
> However, it's worth noting that while the classification loss is vital, the reconstruction loss also holds its significance. Without the reconstruction loss, the model seems to disproportionately focus on the target attribute words, often at the cost of content preservation. This would result in sentences that might have the desired style but lack meaningful content or context.
>
> In essence, the interplay between the classification and reconstruction losses ensures that our model can achieve a fine balance between style transfer and content preservation. Thank you for bringing this to our attention. We will include this discussion in our revised paper to provide a more comprehensive understanding of our methodology.
>
> ---
>
> ***Q2**: How crucial are the parallel connections? Do you think that stack connections might be enough to perform compositional text editing?*
>
> **[Response to Q2]**
>
> The selection between parallel and stack connections is not a matter of one being universally superior to the other, but rather it is contingent upon the specific requirements of the multi-attribute TST task in question.
>
> To elucidate, in tasks that necessitate simultaneous generation of outputs spanning various target styles, the parallel connections are pivotal. They enable the model to cater to all target styles at once without being encumbered by the sequential nature that a stacked approach might entail.
>
> On the other hand, when we are confronted with tasks that require compositional text editing, where several attributes need concurrent modification, the stack connections emerge as the optimal choice. The very architecture of stacking connections ensures that the model can systematically edit and refine the output in a layered manner, integrating multiple attributes seamlessly.
>
> Thus, rather than being redundant or mutually exclusive, the parallel and stack connections function in harmony, each addressing distinct facets of multi-attribute TST tasks. Their incorporation ensures that our model remains versatile and aptly equipped to handle a myriad of text style transfer challenges.
>
> We truly appreciate your keen observation, and we will include this discussion in the methodology section of our revised manuscript to convey their collective importance more effectively.
>
> ---
>
> ***Q3**: Have you considered and conducted experiments using other models, in addition to BART-Large?*
>
> **[Response to Q3]**
>
> As mentioned in our earlier response, your point regarding the broader applicability of AdapterTST to models other than BART-Large is well-taken. Our framework is designed to be model-agnostic, emphasizing its versatility. Due to computational constraints and time limitations, we could only include the results for applying AdapterTST on T5. Below table shows the benchmarking results on Yelp dataset.
>
> |Model| ACC| Bertscore| PPL| G-Score|
> |-|-|-|-|-|
> |BackTrans |94.5 |0.88 |11.3 |1.95|
> |CrossAlign |74.3| 0.89| 35.3| 1.23|
> |DeleteOnly |87.6 |0.91 |36.4 |1.30|
> |Del&Retri |90.2| 0.91| 34.0| 1.34|
> |DualRL |88.9| 0.95| 27.1 |1.46|
> |Template |83.7| 0.92| 47.2| 1.18|
> |Unpaired |50.6| 0.91| 53.1| 0.95|
> |UnsuperMT |96.2| 0.93| 33.5| 1.39|
> |Style Transformer |85.8 |0.95 |10.1 |2.00|
> |Adapter-TST_BART-Large| 90.1| 0.91| 8.2| 2.15|
> |Adapter-TST_T5-Large| 97.3|0.89|4.8|2.62 |
>
> We noted that applying Adapter-TST_T5-Large yield the best performance on the Yelp dataset. We will complete the experiments and apply Adapter-TST on other LLMs such as LLaMA, and test the models on multiple stylistic attribute dataset. The updated results will be reported in the camera-ready version of our paper.
>
> ---
>
> ***Q4**: Would your proposed method demonstrate favorable performance if trained on a combination of multiple textual style transfer datasets, such as sentiment, formality, grammar correction, and others?*
>
> **[Response to Q4]**
>
> You've touched upon a salient aspect of our work. Our Adapter-TST is not only constructed with a view towards handling individual style attributes but also thrives in settings where multiple attributes converge. If subjected to an amalgamated dataset that spans sentiment, formality, grammar correction, among others, we are confident that our methodology would manifest commendable performance.
>
> A defining feature of the adapter-based architecture is its innate flexibility and ability to concurrently assimilate diverse attributes. With the deployment of the parallel connections, the model is programmed to allow each adapter to home in on, and specialize in, a particular attribute. This ensures that even in the face of a multifaceted dataset, each adapter can function autonomously and proficiently, leading to adept multi-attribute text style transfers.
>
> Further underscoring our commitment to advancing this domain, we intend to broaden our empirical horizons and embark on a series of experiments which will evaluate Adapter-TST's prowess when trained on combined TST datasets. We will report the experiment results in our revised paper. These findings will certainly shed more light on our model's versatility and robustness.
>
> ---
>
> ***Others**: Missing References*
>
> We will add the suggested reference and perform a more thorough literature review in the updated version of our paper.
>
> ---
>
> ***Others**: Grammar Issues*
>
> Thank you for pointing out the grammar mistakes in the paper. We have corrected them, and we will perform a more thorough proofreading in our camera-ready version.

---

### Meta-Review · Area_Chair_xRod · 2023-09-15

**Recommendation:** 3

**Metareview:**

The paper introduces "Adapter-TST," an adapter-based framework for multiple-attribute text style transfer, aiming to address the challenge of fine-tuning large language models with limited data. Adapter-TST utilizes neural adapters and offers parallel and stacked connection configurations, allowing simultaneous control of various stylistic attributes. The proposed method is evaluated on two datasets and compared to baseline models.

**Main Criticisms by Reviewers:**

- **Limited Backbone Models:** Reviewers expressed concerns about the evaluation bias resulting from using only one backbone model (BART-Large). They recommended assessing Adapter-TST with multiple backbone models to ensure generalizability.

- **Lack of Specifics in Human Evaluation:** Reviewers noted the need for more transparency in the human evaluation process, including the number of examples used and the number of workers involved.


The authors have made commendable efforts to address the reviewers' concerns. They provided results for the T5 model to assess generalizability, enhancing the paper's rigor. They also clarified the human evaluation process, offering specific details. Moreover, the authors incorporated newer, relevant baselines for comparison, strengthening their argument for the effectiveness of Adapter-TST.

In summary, the authors have made significant strides in addressing the reviewers' concerns. While some improvements can still be made, their responses demonstrate a commitment to enhancing the paper's quality and addressing potential limitations.

---

### Decision · Program_Chairs · 2023-10-07

**Decision:**

Accept-Findings

**Comment:**

The paper introduces "Adapter-TST," an adapter-based framework for multiple-attribute text style transfer, aiming to address the challenge of fine-tuning large language models with limited data. Adapter-TST utilizes neural adapters and offers parallel and stacked connection configurations, allowing simultaneous control of various stylistic attributes. The proposed method is evaluated on two datasets and compared to baseline models.

**Main Criticisms by Reviewers:**

- **Limited Backbone Models:** Reviewers expressed concerns about the evaluation bias resulting from using only one backbone model (BART-Large). They recommended assessing Adapter-TST with multiple backbone models to ensure generalizability.

- **Lack of Specifics in Human Evaluation:** Reviewers noted the need for more transparency in the human evaluation process, including the number of examples used and the number of workers involved.


The authors have made commendable efforts to address the reviewers' concerns. They provided results for the T5 model to assess generalizability, enhancing the paper's rigor. They also clarified the human evaluation process, offering specific details. Moreover, the authors incorporated newer, relevant baselines for comparison, strengthening their argument for the effectiveness of Adapter-TST.

In summary, the authors have made significant strides in addressing the reviewers' concerns. While some improvements can still be made, their responses demonstrate a commitment to enhancing the paper's quality and addressing potential limitations.